# Early Detection of the Start of the Influenza Epidemic Using Surveillance Systems in Catalonia (PREVIGrip Study)

**DOI:** 10.3390/ijerph192417048

**Published:** 2022-12-19

**Authors:** M. Rosa Dalmau Llorca, Elisabet Castro Blanco, Carina Aguilar Martín, Noèlia Carrasco-Querol, Zojaina Hernández Rojas, Alessandra Queiroga Gonçalves, José Fernández-Sáez

**Affiliations:** 1Primary Care Intervention Evaluation Research Group (GAVINA Research Group), IDIAPJGol Terres de l’Ebre, 43500 Tortosa, Spain; 2Servei d’Atenció Primària Terres de l’Ebre, Institut Català de la Salut, 43500 Tortosa, Spain; 3Campus Terres de l’Ebre, Universitat Rovira i Virgili, 43500 Tortosa, Spain; 4Terres de l’Ebre Research Support Unit, Foundation University Institute for Primary Health Care Research Jordi Gol i Gurina (IDIAPJGol), 43500 Tortosa, Spain; 5Unitat d’Avaluació, Direcció d’Atenció Primària Terres de l’Ebre, Institut Català de la Salut, 43500 Tortosa, Spain; 6Unitat de Recerca, Gerència Territorial Terres de l’Ebre, Institut Català de la Salut, 43500 Tortosa, Spain; 7Unitat Docent de Medicina de Familia i Comunitària, Tortosa-Terres de l’Ebre, Institut Català de la Salut, 43500 Tortosa, Spain

**Keywords:** influenza, public health surveillance, epidemics

## Abstract

Sentinel physician networks are the method of influenza surveillance recommended by the World Health Organization. Weekly clinical diagnoses based on clinical history are a surveillance method that provides more immediate information. The objective of this study is to evaluate which influenza surveillance system is capable of the earliest detection of the start of the annual influenza epidemic. We carried out an ecological time-series study based on influenza data from the population of Catalonia from the 2010–2011 to the 2018–2019 seasons. Rates of clinical diagnoses and of confirmed cases in Catalonia were used to study the changes in trends in the different surveillance systems, the differences in area and time lag between the curves of the different surveillance systems using Joinpoint regression, Simpson’s 1/3 method and cross-correlation, respectively. In general, changes in the trend of the curves were detected before the beginning of the epidemic in most seasons, using the rates for the complete seasons and the pre-epidemic rates. No time lag was observed between clinical diagnoses and the total confirmed cases. Therefore, clinical diagnoses in Primary Care could be a useful tool for early detection of the start of influenza epidemics in Catalonia.

## 1. Introduction

Influenza is a respiratory disease that poses a challenge to Public Health because it has a high pandemic potential, high morbidity, a great capacity to spread, and a high frequency of complications and mortality in risk groups. Influenza is responsible for annual seasonal epidemics, during which an estimated 290,000 to 650,000 people die worldwide [1].

The World Health Organization (WHO) has recommended surveillance of influenza at the global and European levels, with the aim of obtaining data that would allow us to identify the types of circulating viruses and their characteristics, and to detect the appearance of influenza epidemics early on [2].

During the years between the two pandemic periods (influenza A during the 2009–2010 season and SARS-CoV-2 during 2020) the maximum incidence of influenza in Spain ranged between 195.17 and 348.10 cases per 100,000 inhabitants per year [3].

Influenza surveillance in Spain is coordinated by the Spanish Influenza Surveillance System (SISS; Sistema de Vigilancia de la Gripe en España), which collects data provided by networks of sentinel doctors from each of the autonomous communities [4].

Before the SARS-CoV-2 pandemic, each sentinel network was made up of volunteer family doctors and paediatricians from Primary Care, who systematically collected samples from patients with symptoms compatible with influenza and sent them for laboratory confirmation. The population covered by this network was statistically representative at the demographic and geographic levels. The SISS also collected non-sentinel information, comprising all cases confirmed outside the network, such as those confirmed in hospitals and laboratories [4].

Catalonia has reported data from its network of sentinel doctors since 2005. The information is gathered in the Information Plan for Acute Respiratory Infections in Catalonia (PIDIRAC; Plan de Información de las Infecciones Respiratorias Agudas en Cataluña). The PIDIRAC provides up-to-date information on daily morbidity, frequency, severity and epidemic potential of acute respiratory diseases, such as influenza, respiratory syncytial virus, parainfluenza, adenovirus, coronavirus, rhinovirus and enterovirus. This system published the data on the Thursday after the week of registration [5].

As part of the Catalan Institute of Health’s (CIH) Primary Care, the clinical history of the patients is registered in a computer program of clinical records, called eCAP (Estació Clínica d’Atenció Primària), which covers more than 80% of the Catalan population [6]. During the 2009 Influenza A pandemic, the Primary Care Services Information System (Sistema de Información de los Servicios de Atención Primaria; SISAP) developed the DiagnostiCat web tool. This tool collects diagnostic data about nine notifiable diseases, including influenza. In Primary Care, influenza is diagnosed based on the symptoms presented by the patient. All the data published on the DiagnostiCat website are extracted from the eCAP program. They are openly accessible via the website and are published the same week as the diagnosis [6].

Currently, as a result of the SARS-CoV-2 pandemic, the surveillance systems are being reorganized in accordance with the guidelines of the WHO [7] and, in turn, of the Ministry of Health in order to expand the sentinel network and to incorporate SARS-CoV-2 surveillance [8,9] This will enable the surveillance systems to provide information on Acute Respiratory Infections and Serious Acute Respiratory Infections.

The SIVIC website (Sistema d’Informació per a la Vigilància d’Infeccions a Catalunya) [10] has superseded and collated the data from DiagnostiCat [10] and the COVID Data website [11] (open access website for COVID data for Catalonia). It also incorporates confirmatory data from the sentinel physician network. This network has been reorganized and improved to comply with the WHO recommendations. It currently comprises 33 primary care teams from all over Catalonia, which collect 330 samples per week (10 samples per team per week) from patients with symptoms of acute respiratory infection and send them for laboratory confirmation.

Worldwide, several studies have shown that there is concordance [12,13] or correlation [14,15,16,17] between a variety of influenza surveillance systems. Studies of our environment have shown that cases of clinical diagnosis concur with the confirmed cases of influenza in the Balearic Islands and Catalonia [12,13].

Once the concordance between the surveillance systems of Catalonia has been established [12] it would be useful to know whether there is a time lag between them and which of the systems provides the earliest information to detect the start of the annual epidemic as soon as possible.

Attempts to measure the time lag in Portugal and the Netherlands revealed no lag between clinical diagnoses of influenza and confirmed cases [18,19]. A study carried out in Italy indicated that the influenza symptom online notification system detected the peak one week before the local sentinel network [20].

On the other hand, some studies have shown that Joinpoint regressions are useful for retrospectively determining changes in the trends in several pathologies. These may be related to the evolution of the disease or other associated factors [21,22].

If there were no time lag between the three surveillance systems in Catalonia, and allowing for the concordance of the three surveillance systems (DiagnostiCat, sentinel-confirmed cases and total confirmed cases) [12] we would be able to select the most accessible and immediate source to predict the start of the annual influenza epidemic in Catalonia and to optimize its management at the level of public health. The main objective of the study was to assess which surveillance system is capable first of detecting the change in trend in the rate curves that determine the start of the influenza epidemic. Likewise, the time lag between the curves of the surveillance systems and the magnitude of excess diagnostic rates and the surveillance systems were studied.

## 2. Materials and Methods

### 2.1. Design and Study Population

We planned an ecological study of population-based time series, using data from clinical diagnoses of influenza, confirmed by the network of sentinel doctors, and influenza cases confirmed inside and outside the network of sentinel doctors between the 2010–2011 and 2018–2019 influenza seasons. The reference population is that of Catalonia, Spain. Each influenza season is taken as running from week 40 of one year to week 20 of the next.

### 2.2. Data Collection

The data collected are secondary and public and consisted of the following variables: rates of clinical diagnoses, sentinel-confirmed cases and total confirmed cases (Table 1).

The number of weekly clinical diagnoses were taken from the DiagnostiCat website (Table 1), which includes the population attended in any primary care centre using the eCAP [6].

The sentinel-confirmed influenza cases were obtained from PIDIRAC [5].

The total confirmed cases of influenza are calculated from sentinel and non-sentinel information [6] where non-sentinel information refers to those cases of influenza confirmed outside the network of sentinel doctors (hospitals and centres that perform confirmatory laboratory tests) [4] The total numbers of confirmed influenza cases were taken from the website of the Spanish Influenza Surveillance System [12].

### 2.3. Statistics Analysis

#### 2.3.1. Analysis of Trends between the Three Surveillance Systems for Complete Seasons and in the Pre-Epidemic Data

Joinpoint regression models [23] were developed with the Joinpoint regression program (version 4.8.0.1, US National Cancer Institute’s Surveillance Research Program, Bethesda, MD, USA) of the US National Cancer Institute’s Surveillance Research Program [24] Rate data from the three surveillance systems (clinical diagnoses, confirmed by the network of sentinel physicians, total confirmed) of all the seasons studied and pre-epidemic rates were used, determining the start of the epidemic using the Moving Epidemic Method (MEM) [25] from the sentinel network rates (reference surveillance system). The change in trend was studied with pre-epidemic clinical diagnostic data. The MEM determines four thresholds, based on influenza rates from at least five previous seasons, thereby defining five levels of intensity: baseline, low, medium, high and very high. The influenza epidemic was considered to have begun when the epidemic threshold., i.e., that separating the baseline and low-intensity levels, was exceeded [25].

#### 2.3.2. Excess Diagnostic Rates Compared with Other Surveillance Systems

To quantify the excess diagnostic rates with respect to other surveillance systems, the areas under the curve of each surveillance system were calculated using Simpson’s 1/3 method. To calculate the area between the curves of the different surveillance systems before the start of the epidemic, the area of the other surveillance systems was subtracted from the area of the respective clinical diagnosis curves.

#### 2.3.3. Time Lag between Clinical and Confirmed Diagnoses

The cross-correlation function (CCF) was used to evaluate the time lag between clinical diagnosis and influenza confirmations. The CCF allows the time lag between two time series to be determined, it being a measure of the similarity between two curves that allows the correlation between a series at a given time and another series at a later or earlier time. In the graphical representation, the correlation is significant when the significance interval is exceeded.

The curve of the rates of clinical diagnoses compared with that of the rate of confirmed cases (considered the gold standard) may be advanced by i weeks (lag − i) or delayed by i weeks (lag + i).

To evaluate the autocorrelation of the series and make the appropriate corrections, an autoregressive integrated moving average (ARIMA) model with the best fit to the independent time series was derived using the pre-whitening technique. Cross-correlation values were then estimated. This analysis was carried out for the study seasons combined and separately.

Statistical analyses were performed with R (version 4.0.2, R core team, Vienna, Austria) [26].

## 3. Results

### 3.1. Trend Analysis

In the study of the complete seasons, the change in trend in the curve of clinical diagnoses was earlier than the start of the influenza epidemic in seven of the nine seasons (2011–2012, 2013–2014, 2014–2015, 2015–2016, 2016–2017, 2017–2018, 2018–2019). In the 2010–2011 season, the change in trend was detected the same week as the start of the epidemic. In the 2012–2013 season, the change in trend in the curve of clinical diagnoses was detected 3 weeks after the start of the epidemic (Table 2).

The change in trend in the sentinel-confirmed rate curve was towards an earlier time than the start of the epidemic in all the seasons studied, except for the 2012/2013 season, which was detected two weeks after the start of the epidemic. For total confirmed cases, the change in trend was detected before the start of the epidemic in most seasons, except in the 2012/2013 season, when it was detected 2 weeks after the start of the epidemic and in the 2017/2018 season, when the change in trend occurred in the same week as the influenza epidemic began (Table 2).

The week the epidemic started, calculated by the MEM from the clinical diagnostic data, was delayed with respect to the beginning of the reference epidemic in the first four seasons (2010–2011, 2011–2012, 2012–2013, 2013–2014) but was detected early or at the same time as the reference system in the five seasons from 2014–2015 to 2018–2019 (Table 2).

Figure 1 shows the week of the change in trend in the pre-epidemic data of clinical diagnoses versus the beginning of the epidemic calculated from the sentinel-confirmed case rates. A significant change in trend was detected in eight of the nine seasons, the 2018–2019 season being the exception. The change in the trend in clinical diagnosis rates was 1 to 7 weeks earlier than the start of the epidemic (Figure 1).

### 3.2. Quantification of Excess Diagnostic Rates Compared with Other Surveillance Systems

There was a smaller area between the pre-epidemic curves for clinical diagnosis rates and sentinel-confirmed rates during the earlier seasons (from 2010–2011 to 2013/2014, and 2015–2016). However, during the later seasons (2014–2015 and from 2016/2017 to 2018–2019) the area was smaller between the rates of total clinical and confirmed diagnoses (Figure 2).

### 3.3. Time Lag between Surveillance Systems

As an overall result of the evaluation of the seasons between 2010–2011 and 2018–2019, after applying the pre-whitening technique, it was observed that the clinical diagnosis could occur 1 week earlier or later than the sentinel-confirmed cases (lag − 1 CCF = 0.149; lag + 1 CCF = 0.163). However, the clinical diagnoses with respect to the total confirmed cases of influenza do not have a time lag (lag 0 CCF = 0.401) (Table 3 and Figure 3).

Considering now the clinical diagnoses and sentinel-confirmed cases for each season separately, we observed that in five of the nine the cases of clinical diagnosis were between 1 and 3 weeks ahead of the sentinel-confirmed cases and that in another no lag was observed. In two of the seasons (2015–2016 and 2017–2018), clinically diagnosed cases were 1 week ahead of sentinel-confirmed cases. Clinical diagnoses were 2 weeks ahead of sentinel-confirmed in two other seasons (2010–2011 and 2014–2015), and 3 weeks ahead in the 2013–2014 season. In the 2018–2019 season, no time lag was observed between clinical diagnoses and sentinel-confirmed cases. Clinical diagnoses lagged 1 week behind sentinel-confirmed case rates in the 2012–2013 and 2016–2017 seasons (Table 3 and Figure 3).

Clinical diagnoses and total confirmed cases for the different seasons exhibited no time lag in six of the nine seasons studied (2010–2011, 2011–2012, 2015–2016, 2016–2017, 2017–2018, 2018–2019). Of the other three seasons, clinical diagnoses were 3 weeks ahead of the total confirmed cases in two (2013–2014 and 2014–2015) and 1 week behind the total confirmed cases in one season (Table 3 and Figure 3).

## 4. Discussion

This study compares three surveillance systems with the aim of determining which of them is most capable of detecting a change in trend before the start of an influenza epidemic. The analysis of the complete curves by Joinpoint regression detected a significant trend change before or in the same week as the start of the influenza epidemic in the three surveillance systems in eight of the nine seasons studied. The exception was the 2012/2013 season, in which it was detected after the start of the epidemic. The change in trend at the start of the influenza epidemic was detected earlier by the rate of clinical diagnoses in the pre-epidemic period. The exclusive use of pre-epidemic data could be key, since it seems that the Joinpoint regression detects trend changes earlier. Thus, using pre-epidemic data could be valuable for the prospective validation of the start of the epidemic.

The start of the influenza epidemic according to clinical diagnoses was detected before or at the same time as the start according to the reference system from the 2014/2015 season onwards. This advance in clinical diagnoses could provide an early warning of when the epidemic will begin.

As in our study, several authors have used Joinpoint regression to describe retrospectively the trend changes in the development of various diseases. They focused on what happens during the period studied and did not make causal inferences; they identified the beginning of the change but did not further investigate the reasons for them [21,27,28]. Future studies will determine the association of different factors by which such changes in trend can be explained.

Regarding the study to quantify the excess diagnostic rates compared with other surveillance systems, the area between the surveillance curves was smaller during the earlier seasons of the study between clinical diagnoses and sentinel-confirmed cases, but was smaller between clinical diagnoses and total confirmed cases during the later seasons. The smaller the area between the two curves, the greater the agreement of the two systems. Closer agreement was found between the surveillance systems in these same seasons [12].

Turning to the study of the time lag between surveillance systems, globally, the clinical diagnosis curve was 1 week ahead or behind the curve of sentinel-confirmed case rates, but there was no time lag between the curves of the clinical diagnosis and the total confirmed cases. The possible delay or advance of clinical diagnoses relative to sentinel-confirmed cases could be due to small fluctuations in the number of sentinel doctors. There were relatively few of them so the number of positive cases they reported might have affected the representativeness of the sample. However, the total number of confirmed cases of influenza includes all the confirmed cases in Catalonia (based on sentinel and non-sentinel information) and is closer to the curve of clinical diagnoses, with respect to the number of cases and the weeks of the increase, peak and decrease, and no time lag was observed in most of the seasons studied. The absence of a time lag between the total confirmed cases of influenza and the clinical diagnoses, and the good concordance between them [12] means that the use of clinical diagnoses could enable more rapid surveillance because diagnostic clinical data are published 4 days earlier.

At the beginning of the epidemic, the curve of clinical diagnoses begins to rise weeks before the curve of confirmed cases, while both curves dip at the same time when the epidemic ends. This pattern of clinical diagnoses with respect to confirmed cases is not captured by the CCF, which is only detected if the peaks of the curves occur simultaneously.

In the individual analysis of the time lag in the separate seasons, the clinical diagnoses were earlier in most (five of the nine) of the seasons than those confirmed by the network of sentinel doctors. By contrast, with respect to the total number of confirmed cases, no time lag was observed in six of the nine seasons studied.

Several studies have used CCF to assess the time lag between different sources of surveillance data for influenza and other diseases. In a study carried out in Portugal, consultations for influenza were evaluated against cases confirmed by the network of sentinel doctors, revealing that there was no time lag or that consultations for influenza were delayed by 1 week relative to confirmations [18]. Other authors have suggested that Influweb (a voluntary system for online reporting of influenza symptoms) is 1 week behind the suspected cases reported by sentinel physicians in Italy [20]. Another study found no time lag between confirmed cases and diagnoses reported by volunteer physicians covering a representative 2% of the Belgian population [19]. In the case of COVID-19, traffic to a website providing information on symptoms was 3 days ahead of the appearance of cases in China and 19 days ahead of cases in the USA [29].

One of the limitations of this study is that the data used are secondary, so it is not possible to know how many of the DiagnostiCat cases have been confirmed. The data of the confirmed cases were obtained with the WebPlotDigitizer tool. Several studies have verified the validity of the data extracted using this tool [30,31,32].

Currently, and as a result of the SARS-CoV-2 pandemic, the surveillance system for respiratory diseases has been improved by increasing the number of confirmation tests [8,9]. This could improve the agreement and time lag between clinical diagnoses and confirmed cases.

This reorganization in Catalonia has led to a considerable increase in the number of sentinel doctors are involved in surveillance. Whereas there were initially around 59 sentinel doctors who collected a maximum of two weekly samples each, now there are 33 primary care teams that collect 10 weekly samples each and in which various doctors participate in collecting samples. This change will help ensure that virological surveillance of influenza and other respiratory viruses is not weakened during the last and first weeks of the year due to medical staff being on vacation.

Clinical diagnostic data are published 4 days before the report of confirmed cases. In addition, diagnostic data can easily be obtained via the Internet. The new SIVIC website brings together the information on clinical diagnoses and the sentinel physician network, making all the data more accessible [10]. This opens up the possibility of validating the results obtained with the new data generated by the new surveillance system.

SIVIC represents an improvement in the information about clinical diagnoses, breaking them down by sex, age, basic health area and several diagnoses (COVID-19, influenza, pneumonia and bronchiolitis). In the case of confirmed cases, it means that these data can be downloaded easily and retrospectively. It also provides data on the respiratory syncytial virus and COVID-19, and vaccinations against influenza and COVID-19.

Since there is no time lag between the curves and there is concordance [12] between them, the changes in the trend in the pre-epidemic curves have been studied, confirming that the rates of clinical diagnoses begin to increase significantly before the increase in the rates of sentinel-confirmed cases.

## 5. Conclusions

The trend changes in clinical diagnoses precede the sentinel-confirmed diagnoses. The magnitude of the excess clinical diagnoses with respect to confirmed cases has been quantified. In addition, there is no time lag between the curves of the surveillance systems. These findings, together with the proven concordance between total clinical and confirmed diagnoses, suggest that Primary Care clinical diagnoses could be a tool worth considering for the early detection of the start of the seasonal influenza epidemic in Catalonia and for optimizing health system management.

## Figures and Tables

**Figure 1 ijerph-19-17048-f001:**
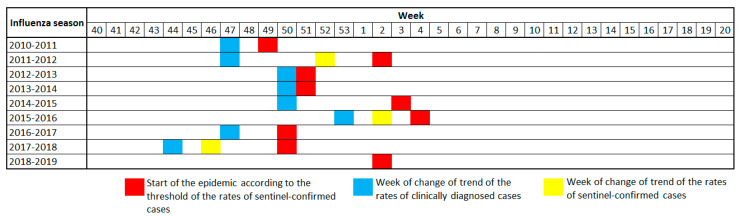
Week of change of trend in pre-epidemic data of clinical diagnoses compared with the beginning of the epidemic, calculated from sentinel-confirmed case rates.

**Figure 2 ijerph-19-17048-f002:**
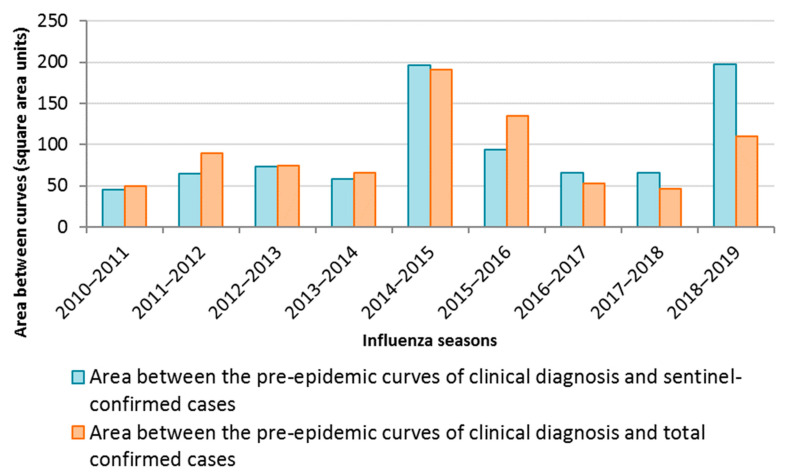
Areas between the pre-epidemic curves of clinical diagnostic, sentinel-confirmed and total confirmed case rates. Start of the epidemic calculated from the rates of sentinel-confirmed cases.

**Figure 3 ijerph-19-17048-f003:**
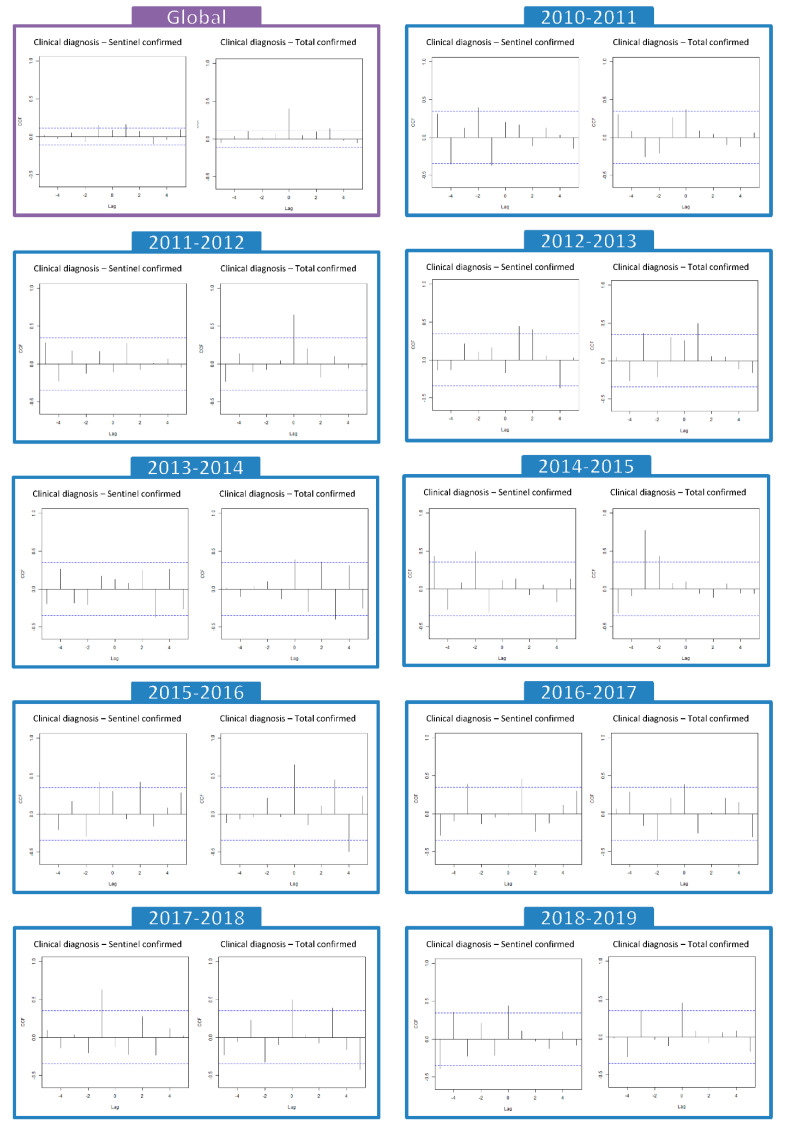
Time lag analysis between clinical diagnoses rate and confirmed cases rate for the global of influenza seasons and for each influenza season. Cross-correlation results. CCF: Cross-correlation Function. --- Significance interval. CCF: ±1 (perfect correlation) with 0 being no correlation. Maximum CCF is positive and lag − i: diagnoses are i weeks ahead of confirmed diagnoses. Maximum CCF is positive and lag i: diagnoses are i weeks behind confirmed diagnoses. Maximum CCF is negative and lag − i: diagnoses lag i weeks behind confirmed diagnoses. Maximum CCF value is negative and lag i: diagnoses are i weeks ahead of confirmed diagnoses.

**Table 1 ijerph-19-17048-t001:** Description of the data sources and variables.

Variable	Clinical Diagnosis	Sentinel-Confirmed	Total Confirmed
Source	DiagnostiCat, now SIVIC	PIDIRAC	SISS
Source Description	Open public website. Secondary data. Weekly update. Published on Sunday of the registration week. https://sivic.salut.gencat.cat/ (accessed on 9 September 2022)	Public website. Secondary data. Weekly update. Published on Thursday following week of registration. https://canalsalut.gencat.cat/ca/professionals/vigilancia-epidemiologica/pla-dinformacio-de-les-infeccions-respiratories-agudes-a-catalunya-pidirac/ (accessed on 15 September 2022)	Public website. Secondary data. Weekly update. Published on Thursday following week of registration. http://vgripe.isciii.es/PresentarGraficos.do (accessed on 15 September 2022)
Method of acquisition	Download from website	WebPlotDigitizer * https://apps.automeris.io/wpd/ (accessed on 15 September 2022)	WebPlotDigitizer * https://apps.automeris.io/wpd/ (accessed on 15 September 2022)
Origin of data	Primary Care of Catalonia, from the Institut Català de la Salut (ICS)	Primary Care sentinel doctors of Catalonia	Hospitals and centres that carry out confirmatory laboratory diagnostic tests in Catalonia
Description of data	Public clinical diagnosis data from SISCAT and other health services in Catalonia that use eCAP (all the ages and sex are included)	Cases confirmed by the network of sentinel doctors in Catalonia, which publishes weekly reports in the Pla d’Informació d’Infections Respiratòries Agudes in Catalonia (all the ages and sex are included) Laboratory confirmation by PCR	Cases confirmed by sentinel doctors and all cases confirmed outside the sentinel network (non-sentinel) (all the ages and sex are included) Laboratory confirmation by PCR
Calculation of rates	$\frac{\begin{array}{c} Number\;of\;clinically\\ diagnosed\;cases \end{array}}{\begin{array}{c} Size\;of\;population\;\\ attended\;annually\\ by\;ICS\; \end{array}}\times 100,000$	$\frac{\begin{array}{c} Number\;of\;\\ sentinel-confirmed\;cases \end{array}}{\begin{array}{c} Size\;of\;population\;assigned\;to\;\\ centinal\;doctor\;anually \end{array}}\times 100,000$	$\frac{\begin{array}{c} Total\;number\;of\;\\ confirmed\;cases\; \end{array}}{\begin{array}{c} Annual\;popopulation\;size,\;\\ Catalonia \end{array}}\times 1,000,000$

Abbreviations. SISCAT: Sistema sanitari integral d’utilització pública de salut de Catalunya (Comprehensive health system for public health use in Catalonia). eCAP: Estación Clínica de Atención Primaria (Primary Care Clinical Station). PCR: Polymerase Chain Reaction. * WebPlotDigitizer: web application that allows extraction of numerical data from a graph. All influenza seasons are included in the three surveillance systems.

**Table 2 ijerph-19-17048-t002:** Result of the Joinpoint analysis of the rate curves for clinical diagnoses, sentinel-confirmed and total confirmed for complete seasons.

Influenza Season	Starting Week of Epidemic. Threshold: MEM Sentinel-Confirmed Reference	Starting Week of Epidemic. Threshold: MEM Clinical Diagnosis	Week Change Trend Clinical Diagnosis	Week Change Trend Sentinel-Confirmed	Week Change of Trend Total Confirmed
2010–2011	49/2010	50/2010	49/2010	47/2010	46/2010
2011–2012	2/2012	4/2012	1/2012	51/2011	1/2012
2012–2013	51/2012	2/2013	2/2013	1/2013	1/2013
2013–2014	51/2013	52/2013	50/2013	48/2013	50/2013
2014–2015	3/2015	52/2014	49/2014	1/2015	1/2015
2015–2016	4/2016	3/2015	1/2016	2/2016	2/2016
2016–2017	50/2016	50/2015	47/2016	48/2016	49/2016
2017–2018	50/2017	50/2017	48/2017	48/2017	50/2017
2018–2019	2/2019	52/2018	50/2018	46/2018	52/2018

**Table 3 ijerph-19-17048-t003:** Cross-correlation function estimates.

		Time Lag (CCF Value)
Influenza Season	Comparison	Pre-Whitened CCF
2010–2011	Clinical diagnoses—Sentinel-confirmed	−2 (0.395)
	Clinical diagnoses—Total confirmed	0 (0.372)
2011–2012	Clinical diagnoses—Sentinel-confirmed	NS
	Clinical diagnoses—Total confirmed	0 (0.649)
2012–2013	Clinical diagnoses—Sentinel-confirmed	1 (0.443)
	Clinical diagnoses—Total confirmed	1 (0.495)
2013–2014	Clinical diagnoses—Sentinel-confirmed	3 (−0.375)
	Clinical diagnoses—Total confirmed	3 (−0.398)
2014–2015	Clinical diagnoses—Sentinel-confirmed	−2 (0.488)
	Clinical diagnoses—Total confirmed	−3 (0.772)
2015–2016	Clinical diagnoses—Sentinel-confirmed	−1 (0.421)
	Clinical diagnoses—Total confirmed	0 (0.650)
2016–2017	Clinical diagnoses—Sentinel-confirmed	1 (0.460)
	Clinical diagnoses—Total confirmed	0 (0.388)
2017–2018	Clinical diagnoses—Sentinel-confirmed	−1 (0.628)
	Clinical diagnoses—Total confirmed	0 (0.497)
2018–2019	Clinical diagnoses—Sentinel-confirmed	0 (0.440)
	Clinical diagnoses—Total confirmed	0 (0.452)
Global	Clinical diagnoses—Sentinel-confirmed	1 (0.163)/−1 (0.149)
	Clinical diagnoses—Total confirmed	0 (0.401)

NS: no significant results; CCF: cross-correlation function.

## Data Availability

The data are secondary and public, DiagnostiCat collects the clinical diagnoses of all primary care physicians who use the eCAP [10]. The data on sentinel-confirmed cases of influenza were obtained from the network of sentinel physicians of Catalonia, which publishes them in the Pla d’Informació de les Infeccions Respiratòries Agudes a Cataluña (PIDIRAC) [33]. The data on total confirmed cases of influenza were extracted from the Spanish Influenza Surveillance System website [34].

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
