# Peer review of "Early Detection of the Start of the Influenza Epidemic Using Surveillance Systems in Catalonia (PREVIGrip Study)"

_ijerph, 2022, doi:10.3390/ijerph192417048_

Round 1

Reviewer 1 Report

I have just curious how authors define influenza season in the data. Does all three data  have laboratory result of influenza? if so, which test do they perform in each dataset.

Figure 2 is hard to understand. Authors need to give y axis title and check legend again for better understanding

Author Response

Dear reviewer, thank you for your comments, they will be very usefull to improve our MS.

I have just curious how authors define influenza season in the data. Does all three data  have laboratory result of influenza? if so, which test do they perform in each dataset.

Each influenza season is taken as running from week 40 of one year to week 20 of the next. This information is specified in Materials and methods, in the last line of the study design.

The data that have laboratory confirmations are sentinel confirmed cases and total confirmed cases, and this confirmation is done by PCR (Polymerase Chain Reaction). Clinical diagnostic data are data extracted from medical records and are not laboratory confirmed. We have added the laboratory test used in table 1.

Figure 2 is hard to understand. Authors need to give y axis title and check legend again for better understanding

Thanks, we have added the titles of the axes to the figure and revised the legend.

Reviewer 2 Report

The manuscript entitled “Early detection of the start of the influenza epidemic using 2 surveillance systems in Catalonia (PREVIGrip Study)”, the authors describe the influenza surveillance system that is capable of the earliest detection of the start of the annual influenza epidemic. The studies are interesting, well designed, and well accepted, minor edition need to consider that may improve the manuscript quality:

-The patient demographic data (age, sex, seasons of clinical history) and a descriptive epidemiologic analysis would be great help. Data as supplement is fine.

- Results section: Confirmed case based on what? Molecular or serological?

- Figure 3: Figure legend should be included to better understand the Cross-correlation. The main message from the figure 3 should be in the legend.

- At introduction, missing of spaces gap specially after reference citation

Author Response

Dear reviewer, thank you for your comments, they will be very usefull to improve our MS.

The manuscript entitled “Early detection of the start of the influenza epidemic using 2 surveillance systems in Catalonia (PREVIGrip Study)”, the authors describe the influenza surveillance system that is capable of the earliest detection of the start of the annual influenza epidemic. The studies are interesting, well designed, and well accepted, minor edition need to consider that may improve the manuscript quality:

-The patient demographic data (age, sex, seasons of clinical history) and a descriptive epidemiologic analysis would be great help. Data as supplement is fine.

The data used in this study are population-based and therefore include people of all ages and both sexes. We have included this information in Table 1 (Material and methods).

- Results section: Confirmed case based on what? Molecular or serological?

Cases are laboratory confirmed by molecular methods (PCR, Polymerase Chain Reaction). We have included this information in Table 1 (Material and methods).

- Figure 3: Figure legend should be included to better understand the Cross-correlation. The main message from the figure 3 should be in the legend.

Thanks, we have modified the legend in figure 3 for easier understanding.

- At introduction, missing of spaces gap specially after reference citation

Thanks, we have revised the entire manuscript and added the missing spaces.

Round 2

Reviewer 1 Report

I have no further comments on this manuscript